# Implementation or Suppression of the Collective Lasing of the Laser Channels at the Intracavity Spectral Beam Combining

**Vladimir A. Kamynin \*, Vitalii V. Kashin, Dmitri A. Nikolaev and Vladimir B. Tsvetkov**

Prokhorov General Physics Institute of the Russian Academy of Sciences, 38 Vavilov St., 119991 Moscow, Russia; kashin@lsk.gpi.ru (V.V.K.); nikolaev@lsk.gpi.ru (D.A.N.); tsvetkov@lsk.gpi.ru (V.B.T.)
\* Correspondence: kamyninva@gmail.com

**Abstract:** The spectral and spatial output parameters of a two-channel, Yb-doped fiber laser operating in the intracavity spectral beam combining mode were investigated. We showed that by using active media with slightly different gain spectra, it is possible to implement either the spectral combining mode of the independent laser channels or the mode of collective lasing of the channels. The difference in the gain spectra of the active media was realized due to the difference in the threshold inverse population in the Yb-doped fibers.

**Keywords:** multi-channel laser; spectral beam combining; collective lasing; fiber lasers





## 1. Introduction

Beam combining (BC) can realize power scaling of lasers. One of the methods of BC is spectral beam combining (SBC). In one version of this method, intracavity combining of laser channels with different wavelengths is performed. Recent laser SBC systems based on fiber active media reach an output power ranging from a few kW [1–3] to over 30 kW when 96 channels are combined [4].

A number of aspects of SBC during the implementation of a set of lasers being placed in a common cavity have been considered in different publications [5–10]. System efficiency is analyzed with spectral and power restrictions on the number of combined laser channels, inclusion of the influence of an aberration on the lasing parameters, etc. However, each laser oscillator in the set is considered independently. At the same time, the situation is possible when some of the laser channels are involved in common (collective) lasing [11–16]. Such a regime was experimentally realized and theoretically justified in [17]. It was demonstrated that the spectral and spatial (directions of beam propagation) parameters of collective lasing (CL) might be essentially different from the parameters in an SBC mode with independent channels. At the same time, CL can take a significant portion of the power from SBC lasing. In such cases, it can be considered a negative effect.

The purpose of our current study is to investigate the possibility of both suppression and implementation of CL in a laser with an intracavity SBC. A two-channel Yb-fiber laser with different gain spectra was used as a model system. The difference in gain spectra might be demonstrated by the difference in amplified spontaneous emission (ASE) spectra of the different channels. The laser cavity was designed according to a scheme with a common output coupler and a diffraction grating (DG) mounted under a grazing angle of incidence [5–8,10,17] (Littman–Metcalf scheme).

The difference in the gain spectra of Yb-active fiber media can be produced in several ways. For example, it can be caused by the difference in the length of the active medium in different laser channels [18]. In our case, this difference was realized via the difference in reflectivity of the fiber end mirrors. This difference led to a difference in the active media's threshold inverse populations in different laser channels. This, in turn, led to a spectral change in the balance between the processes of absorption and amplification of radiation and to a change in the spectral parameters of the gain (and ASE also) [18].

## 2. Experimental Setup

The basic optical schematics of the experiment and the beams paths in the laser cavity were designed to be similar to [17] and are shown in Figure 1. Two single-mode, silica glass, double-clad Yb-doped fibers, A and B, were used as the active media. During the experiment, fibers A and B could occupy position 1 or position 2 (Figure 1). The core diameter of the fibers was about 6 μm and the numerical aperture was $NA^F = 0.11$. The lengths of the active fibers were practically the same and equal to 2 m. The front-end faces (facing the output mirror, Figure 1) of the fibers were polished under an angle of 10°. The polished fiber ends were fixed in the common holder in such a way that the center-to-center distance of the fibers was $\Delta X \cong 125$ μm, and the axes of the fibers were parallel and settled in the XY plane. The axes of the beams leaving both fibers were parallel to each other and with the cavity axis. Both fibers could move synchronously along the *X*-axis. A fiber ring mirror with a reflectivity of $R_B = 90\%$ was placed on the back of fiber B, but the back facet of fiber A was cleaved ($R_A = 4\%$). For the pumping of the active fibers, we used 976 nm laser diodes.

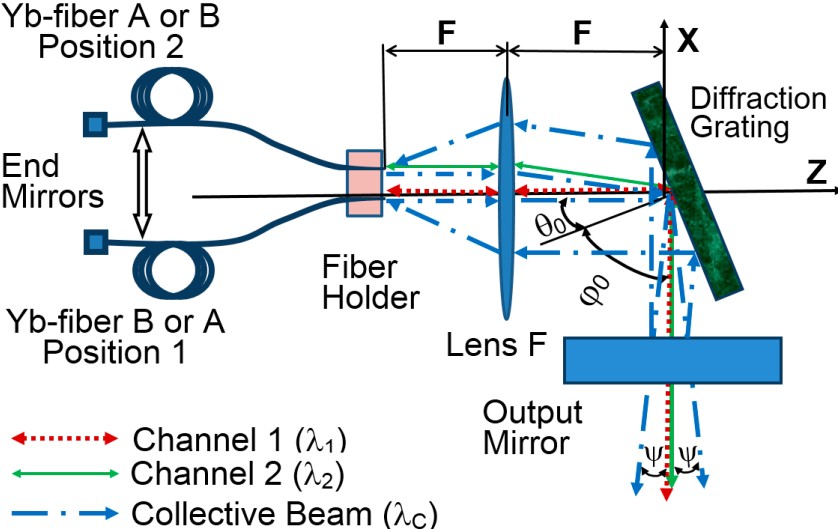

**Figure 1.** The optical scheme of the laser cavity and the schematic of the beam paths. "Beam" means the central ray of the beam.

The threshold pump power of lasing channel 1 with active fiber B (position 1 (X = 0), $\lambda_1 = 1040 \pm 0.4$ nm) was about 0.6 W. The threshold pump power of active fiber A (position 2, $X \cong 125$ μm, $\lambda_2 = 1033.3$ nm) was near 1.4 W. The amplified spontaneous emission spectra of active fibers A and B are shown in Figure 2a. The front ends of the fibers were placed in the focal plane of a positive lens F, with a focal length of F = 55 mm. This lens transformed the initial beams with a full divergence angle of 2 VF ≈ 0.22 rad (corresponding to $NA_F = 0.11$) into quasi-parallel (2 V ≈ $5 \times 10^{-5}$ rad) Gaussian beams with radii of ω ≈ 6.3 mm at the $1/e^2$ level. Furthermore, these beams were reflected by the DG with a grating groove density of N = 300 g/mm. The diffracted beams fell normal on the plane output mirror ($R_{out} = 85\%$ in a spectral range of 1000–1100 nm). The point of intersection of the grating surface with the optical axis of the cavity and the optical axes of the beams (central rays) was located in the right focal plane of lens F. The sum of the angle of incidence $\theta_0$ (an angle between the X-axis and the normal to the DG) and the angle of diffraction $\varphi_0$ (diffraction order equal to −1) became $\theta_0 + \varphi_0 = 90°$ at a lasing wavelength equal to 1040 nm. The values of $\theta_0 = 32.25°$ and $\varphi_0 = 57.75°$ corresponded to these requirements. The diffraction efficiency was about 60%. The distance from the DG to the output mirror was 85 mm.

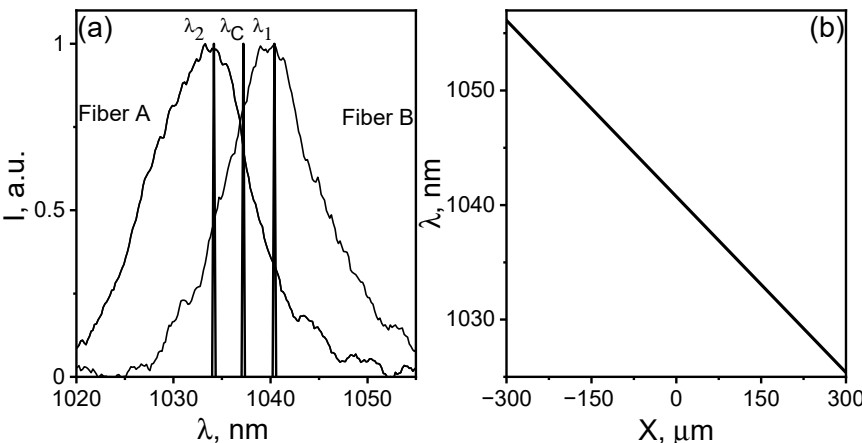

**Figure 2.** (**a**) The ASE spectra of fibers A and B and relative position of the lasing wavelengths of fiber 1 ($\lambda_1$), fiber 2 ($\lambda_2$), and the collective mode ($\lambda_C$). (**b**) Dependence of lasing wavelength on the single-fiber transversal coordinate X.

The spectral parameters of the laser output were studied using a spectrum analyzer ASP-150C (Avesta-Project Inc., Moscow, Russia). The spectral range of the analyzer was 460–1100 nm, and the resolution was equal to 0.4 nm.

## 3. Results

The wavelengths $\lambda_1$ and $\lambda_2$ corresponding to the independent lasing of channels 1 and 2 are determined based on the positions (transverse coordinates $X_i$) of the active fibers and are defined by the following equation [17]:

$$\lambda_{1,2} = d(\sin(\theta_0 + \text{arctg}(X_{1,2}/F)) - \sin\varphi_0)/m_1, \tag{1}$$

where d = 1/N; F = 55 mm, which is the focal length of the positive lens F; and $m_1 = -1$, which is the diffraction order. Figure 2b shows the calculated dependence of the lasing wavelengths on the single-fiber transverse coordinate X.

The Xi coordinates of both active fibers are connected in our case: $X_2 = X_1 + \Delta X$, where $\Delta X = 125$ μm is the distance between the fiber centers. At $X_1 = 0$ and $\lambda_1 \cong 1040$ nm, the calculated value of $\lambda_2$ is approximately 1033.6 nm. The calculated difference between the wavelengths of the laser channels is practically constant in the range $\lambda_1 = 1030 \div 1050$ nm and is within $\Delta\lambda = \lambda_1 - \lambda_2 = 6.39 \div 6.41$ nm. The wavelength of the collective wave $\lambda_C$ should be between the wavelengths $\lambda_1$ and $\lambda_2$ and should be equal to the following [17]:

$$\lambda_c \cong (\lambda_1 + \lambda_2)/2, \tag{2}$$

The calculated value of the collective beam wavelength is equal to 1036.7 nm at $X_1 = 0$. The collective laser emission consists of two beams (Figure 1) that propagate at angles $\pm\psi$ with respect to the normal to the output mirror. These angles are calculated as follows [17]:

$$\psi(\lambda_c) = \varphi_0 \pm \varphi_c(\lambda_c), \tag{3}$$

where

$$\varphi_c(\lambda_c) = \arcsin\left[\sin(\theta_o + \text{arctg}(X_1/F)) - m_1\lambda_c/d\right]. \tag{4}$$

Under our experimental conditions and $X_1 = 0$ and $\lambda_1 \cong 1040$ nm, the inclination angles of the output collective beams are $\psi = \pm 1.8$ mrad.

The results of the experimental studies demonstrate that the lasing spectrum and the propagation direction of the output emission depend on the positions occupied by the active fibers A and B. In the case when position 1 ($X_1 = 0$) is occupied by active fiber B, the lasing spectrum consists of two lines with maximum $\lambda_1 = (1040 \pm 0.4)$ nm and $\lambda_2 = (1033.3 \pm 0.4)$ nm. This spectrum is shown in Figure 3a. These wavelengths are

in good agreement with the calculated values for $\lambda_1$ and $\lambda_2$. The output radiation was realized as a single beam propagating along the resonator axis. This is a spectral beam combining mode.

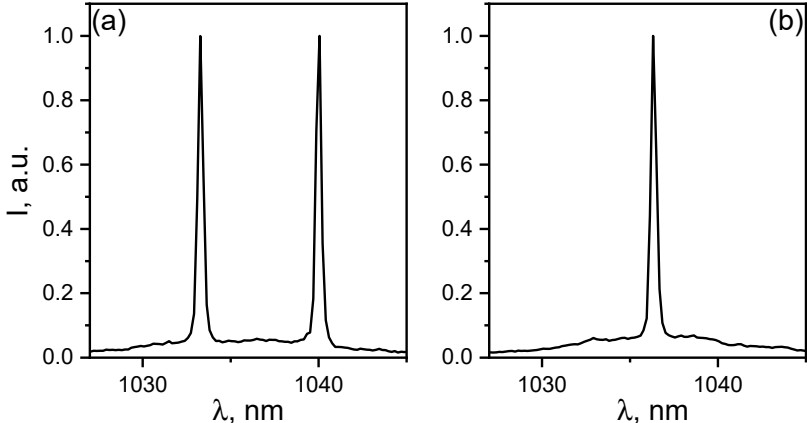

**Figure 3.** Spectra of lasing when (**a**) fiber B is in position 1 while fiber A is in position 2, and (**b**) fiber A is in position 1 while fiber B is in position 2.

In the case where position 1 ($X_1 = 0$) is occupied by fiber A, the spectrum presents a single line with the maximum at $(1036.3 \pm 0.4)$ nm (Figure 3b). This value corresponds to the wavelength of the collective beam. The laser output radiation was realized by two beams propagating at angles of $\pm(1.8 \pm 0.2)$ mrad with respect to the normal to the output mirror. These values are in good agreement with the values predicted by Equation (3). This is collective mode lasing.

The beam quality is the same with the measured value $M^2 \approx 1.2$ for all the beams, both for single-beam and two-beam lasing.

The measured values of the laser linewidth are about 0.4 nm. However, these values are consistent with the instrument function of the spectrum analyzer. The calculated value of linewidth is identical for all lines and is about 0.2 nm.

## 4. Discussion

The presence of two different lasing modes can be explained as follows: When active fiber B is located in position 1 (on the resonator axis, $X_1 = 0$), it is responsible for lasing beam 1 ($\lambda_1 = 1040$ nm) (Figures 1 and 2a). Beam 2 ($\lambda_2 = 1033.3$ nm) is generated by active fiber A. The collective beam ($\lambda_c = 1036.3$ nm) is generated simultaneously by fibers 1 and 2. Therefore, the gain of independent beams 1 and 2 is higher than the collective beam (Figure 2). Therefore, lasing of beams 1 and 2 is preferable when position 1 is occupied by fiber B.

If fiber A is located at position 1, then the situation is reversed and collective beam lasing with wavelength $\lambda_c$ is preferable.

This situation may be illustrated as follows: We can calculate the gain of the collective wave $G_c$ ($\lambda_c$) after a round trip of the collective beam in the cavity. In this case, we assume that the threshold conditions for laser channels 1 (beam 1, $\lambda_1 = \lambda_c + \Delta\lambda/2$, where $\Delta\lambda = \lambda_1 - \lambda_2$) and 2 (beam 2, $\lambda_2 = \lambda_c - \Delta\lambda/2$) are satisfied at any value of $\lambda_c$. At and below this value, we assume that lasing in each of the possible channels (1, 2, and collective) is absent. It should also be noted that the resonator length and the length of the active medium of the collective lasing channel are twice as long as the lengths of the cavities and the active media of channels 1 and 2, respectively.

The threshold radiation gain per pass of the active media A and B (regardless of their position) is $g_{th}^A = 9.04$ and $g_{th}^B = 1.9$. The gain spectra (normalized to their maximum values), $g_A(\lambda)$ and $g_B(\lambda)$, of media A and B in our case are well approximated by the Gaussian function:

$$g_{A,B}(\lambda) = \exp\left[-4\ln2\left(\frac{(\lambda - \lambda_{A,B})^2}{\Delta\lambda_{A,B}{}^2}\right)\right], \tag{5}$$

where $\lambda_A$ = 1033.5 nm and $\lambda_B$ = 1040 nm are the maxima of the spectral lines, and $\Delta\lambda_A$ = 11.5 nm and $\Delta\lambda_B$ = 11.5 nm are their full widths at half maximum. Taking into account the above information, the gain of the collective wave after one round trip in the resonator along its trajectory is equal to

$$G_c^{1A}(\lambda) = g_{th}^{A\,2} g_{th}^{B\,2} \left(\frac{g_A(\lambda)}{g_A(\lambda + \Delta\lambda/2)}\right)^2 \left(\frac{g_B(\lambda)}{g_B(\lambda - \Delta\lambda/2)}\right)^2 R_{out}{}^2 D^4 R_A R_B, \tag{6}$$

when active fiber A is in position 1. Here, $D = 0.6$ is the absolute efficiency of the DG, Rout = 0.85 is the reflectivity of the output mirror, and $R_A$ = 0.04 and $R_B$ = 0.9 are the reflectivity of the back-end mirrors of fibers A and B, respectively.

For the case when fiber A is in position 2, the corresponding collective wave gain is

$$G_c^{2A}(\lambda) = g_{th}^{A\,2} g_{th}^{B\,2} \left(\frac{g_A(\lambda)}{g_A(\lambda - \Delta\lambda/2)}\right)^2 \left(\frac{g_B(\lambda)}{g_B(\lambda + \Delta\lambda/2)}\right)^2 R_{out}{}^2 D^4 R_A R_B, \tag{7}$$

The dependencies are shown in Figure 4 for both cases.

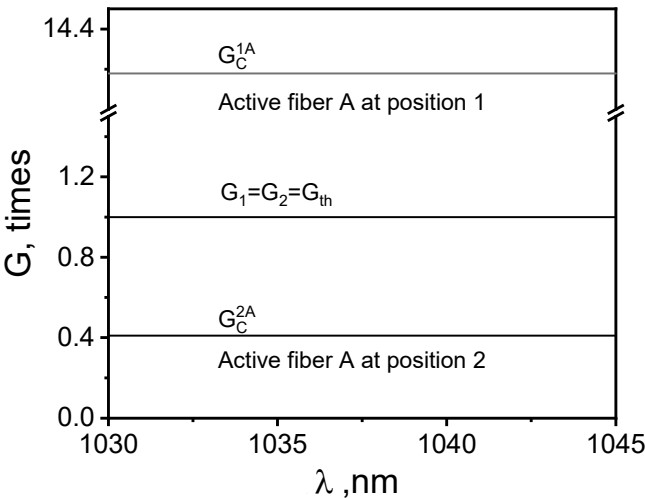

**Figure 4.** Gain of collective lasing radiation $G_c$ after one round trip in the resonator as a function of $\lambda$ at different positions of active fibers. The same gains $G_1$ and $G_2$ of beams 1 and 2 are equal to the threshold value.

As can be seen from the calculation results (Figure 4), the laser operates either in the spectral combining mode of the independent laser channels or in the collective laser channel mode, depending on the positions of the active fibers A and B.

In our case, the values of $G_C^{1A}$ and $G_C^{2A}$ are practically independent of the wavelengths. The results of the analysis show that the dependence of $G_C^{1A}$ and $G_C^{2A}$ on $\lambda$ occurs only at different widths of the gain spectral lines of media A or B. For example, under our conditions and with a possible change in the value of $\Delta\lambda_A$ or $\Delta\lambda_B$ at $\pm0.5$ nm, the changes in both $G_C^{1A}(\lambda)$ and $G_C^{2A}(\lambda)$ reach a twofold value within the considered spectral range of 1030–1045 nm. In the case of a higher difference between $\Delta\lambda_A$ and $\Delta\lambda_B$, in both positions of the fibers, it is possible to achieve either independent generation or collective one.

## 5. Conclusions

An explanation is proposed for the effect of collective lasing in a laser with intracavity spectral beam combining due to the difference in gain spectra in the channels. It is shown that the spectral (lasing wavelengths) and spatial (radiation propagation direction) laser

output parameters differ in the independent-channel lasing mode and in the spectral beam combining mode. This should be considered in the design of multichannel lasers. A method for both suppression and implementation of collective lasing in a laser with intracavity spectral beam combining is proposed and investigated.

**Author Contributions:** Conceptualization, D.A.N. and V.B.T.; methodology, D.A.N.; investigation, D.A.N. and V.A.K.; data analysis, D.A.N.; writing—original draft preparation and editing D.A.N., V.A.K. and V.B.T.; visualization, D.A.N.; resources, D.A.N., V.A.K. and V.V.K.; project administration, V.B.T. All authors have read and agreed to the published version of the manuscript.

**Funding:** This research received financial support from the Ministry of Science and Higher Education of Russian Federation, grant number 075-15-2022-315, and was carried out at the World-Class Research Center «Photonics».

**Institutional Review Board Statement:** Not applicable.

**Informed Consent Statement:** Informed consent was obtained from all subjects involved in the study.

**Data Availability Statement:** The data supporting the findings of this study are available within the article.

**Conflicts of Interest:** The authors declare no conflict of interest.

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
