# Peer review of "Implementation or Suppression of the Collective Lasing of the Laser Channels at the Intracavity Spectral Beam Combining"

_photonics, doi:10.3390/photonics10091022_

Round 1

Reviewer 1 Report

The spectral and spatial output parameters of a two-channel Yb-doped fiber laser operating in the intracavity spectral beam-combining mode were thoroughly examined. Investigation of this article has demonstrated that through the utilization of active media exhibiting marginally distinct gain spectra, it becomes feasible to realize either the spectral combining mode of independent lasing channels or the mode characterized by the collective lasing of the channels. However, there are some elements that remain unclear and need to be modified. The manuscript should be modified and improved. My suggestions are as follows (see list below).

1. In line 44, the HR mirror is used only once in manuscript, and the meaning of HR is still missing. I guess the HR mirror is high reflectivity mirror. Using the high-reflectivity mirror to replace the HR mirror is better than before.

2. In line 72, the meaning of N is missing. To give the definition of N will make your article easier to read.

3. In line 65, the position is contain the information of axis X. Replacing the “position 1, X=0” by “position 1 (X=0)” as you do in the later part of article will be better.

4. In line 140, you need add the special situation such as “the lasing of beam 1 and 2 is preferable when the position 1 is occupied by fiber B” to decrease the misleading of your article.

5. Fig. 1 shows the situation in which position 1 is occupied by beam A and position 2 is occupied by beam B. In line 51, “during the experiment, fibers A and B could occupy position 1 or 2 (Fig. 1)” is unsuitable. Using “during the experiment, fibers A and B could occupy position 1 or 2. Fig. 1 shows the situation that the position 1 is occupied by beam A and position 2 is occupied by beam B” will make your article more rigorous.

6. In recent years, quantum communication, which leverages the security offered by quantum mechanics, has been gaining significant traction. In this context, the application of spectral beam combining is considered a potential technique, especially in the field of quantum information processing and quantum communication. By incorporating pertinent content related to quantum communication, the article can further highlight the practical implications and potential applications of the proposed work. Therefore, I suggest that the authors consider citing the following articles [Nat. Photonics 8, 595–604 (2014), Nat. Photonics 16, 154–161 (2022), Sci. Bull. 67, 2167-2175 (2022), National Science Review 10, nwac228 (2023)] which may make your article be better.

Once the authors have addressed the points raised, I will agree that this work can be published in Photonics.

Moderate editing of English language required

Author Response

Dear Reviewer!

We are deeply grateful to you for the consideration of possibility of publishing of the article in Photonics journal and for your work to improve the quality of the article. 

We take into account all the remarks.

Please find our answers below.

Sincerely yours

Vladimir Kamynin

Comment 1: In line 44, the HR mirror is used only once in manuscript, and the meaning of HR is still missing. I guess the HR mirror is high reflectivity mirror. Using the high-reflectivity mirror to replace the HR mirror is better than before.

A: We have replaced “HR mirrors” with “End mirrors”.

Comment 2: In line 72, the meaning of N is missing. To give the definition of N will make your article easier to read.

A: We made the corrections in the text:

….these beams were reflected by DG with the grating groove density of N = 300 g/mm.

Comment 3: In line 65, the position is contain the information of axis X. Replacing the “position 1, X=0” by “position 1 (X=0)” as you do in the later part of article will be better.

A: We made the corrections in the text:

…” with active fiber B (position 1 (X=0), λ1=1040±0.4 nm)”….

Comment 4: In line 140, you need add the special situation such as “the lasing of beam 1 and 2 is preferable when the position 1 is occupied by fiber B” to decrease the misleading of your article.

A: We made the corrections in the text:

…“lasing of beams 1 and 2 is preferable when the position 1 is occupied by fiber B.”

Comment 5: Fig. 1 shows the situation in which position 1 is occupied by beam A and position 2 is occupied by beam B. In line 51, “during the experiment, fibers A and B could occupy position 1 or 2 (Fig. 1)” is unsuitable. Using “during the experiment, fibers A and B could occupy position 1 or 2. Fig. 1 shows the situation that the position 1 is occupied by beam A and position 2 is occupied by beam B” will make your article more rigorous.

 A: We have replaced Fig.1.

Comment 6: In recent years, quantum communication, which leverages the security offered by quantum mechanics, has been gaining significant traction. In this context, the application of spectral beam combining is considered a potential technique, especially in the field of quantum information processing and quantum communication. By incorporating pertinent content related to quantum communication, the article can further highlight the practical implications and potential applications of the proposed work. Therefore, I suggest that the authors consider citing the following articles [Nat. Photonics 8, 595–604 (2014), Nat. Photonics 16, 154–161 (2022), Sci. Bull. 67, 2167-2175 (2022), National Science Review 10, nwac228 (2023)] which may make your article be better.

A: Thank you for your comment. Our work is dedicated to spectral beam combining method and implementation or suppression of collective lasing in such systems. To the best of our knowledge, SBC is not now used in quantum communication systems. Therefore, to avoid reader confusion, we decided not to include listed articles in our manuscript.

Reviewer 2 Report

The work is devoted to the demonstration and explanation of the effect of collective lasing in a laser with intracavity spectral beam combining due to the difference in the gain spectra in the channels. The subject of this study is of great importance for the development of laser technologies. This study may be of great interest to readers if the quality of the presentation is slightly improved. (1) I would advise the authors to supplement the manuscript with a full-fledged layout of the laser cavity for clarity. It will allow readers to distinguish between the fibers A and B. (2) The authors claim that switching between the spectral beam combining mode and the collective lasing mode can be done by swapping positions of the fibers A and B in the optical arrangement shown in Fig.1. I would suggest the authors to comment if there is any possibility to achieve the similar effect without swapping the fibers’ positions physically, for instance by shifting the optical axis of the lens or by tilting the grating? In my opinion, translation of the optical axis of the lens can be considered similarly to the fibers’ position swapping.  (3) In the Conclusions: The authors’ statement “Since the spectral parameters of collective lasing are different from the spectral parameters of the independent channel lasing, it is possible to realize conditions for either predominant amplification of collective lasing emission or its suppression.” is rather general, not summarizing the findings of the conducted study. The conclusion should summarize the main findings and indicate their fundamental and/or applied significance.

Proofreading and correction of minor spelling errors are recommended. 

For instance, in the line 121:  "prsented"

Author Response

Dear Reviewer!

We are deeply grateful to you for the consideration of possibility of publishing of the article in Photonics journal and for your work to improve the quality of the article. 

We take into account all the remarks.

Please find our answers below.

Sincerely yours

Vladimir Kamynin

1)      I would advise the authors to supplement the manuscript with a full-fledged layout of the laser cavity for clarity. It will allow readers to distinguish between the fibers A and B.

A: We have replaced Fig. 1

2)      The authors claim that switching between the spectral beam combining mode and the collective lasing mode can be done by swapping positions of the fibers A and B in the optical arrangement shown in Fig.1. I would suggest the authors to comment if there is any possibility to achieve the similar effect without swapping the fibers’ positions physically, for instance by shifting the optical axis of the lens or by tilting the grating? In my opinion, translation of the optical axis of the lens can be considered similarly to the fibers’ position swapping.

A: You are right. Changing the sign of the angle of incidence of the radiation on the grating (rotating the grating  at an angle of 2θ0≈64.50 in the plane Fig.1) will lead to the inversion of the laser wavelengths of the channels. However, we found this way too complicated. In this way it is necessary to additionally rotate the grating around the cavity axis (1800).

The translation of the optical axis of the lens will only lead to a synchronous change in the lasing wavelengths in both channels (Fig.2b).

3)      In the Conclusions: The authors’ statement “Since the spectral parameters of collective lasing are different from the spectral parameters of the independent channel lasing, it is possible to realize conditions for either predominant amplification of collective lasing emission or its suppression.” is rather general, not summarizing the findings of the conducted study. The conclusion should summarize the main findings and indicate their fundamental and/or applied significance.

A: We have corrected the conclusions (with taking into account your comments) (see revised version).

Round 2

Reviewer 1 Report

This manuscript can be accepted for publication.

Minor editing of English language required